# Hyaluronidase-Responsive Mesoporous Silica Nanoparticles with Dual-Imaging and Dual-Target Function

**DOI:** 10.3390/cancers11050697

**Published:** 2019-05-20

**Authors:** Zhi-Yuan Wu, Cheng-Chang Lee, Hsiu-Mei Lin

**Affiliations:** 1Department of Bioscience and Biotechnology, National Taiwan Ocean University, Keelung City 20224, Taiwan; j18185202000@yahoo.com.tw (Z.-Y.W.); d91051238@gmail.com (C.-C.L.); 2Center of Excellence for the Oceans, National Taiwan Ocean University, Keelung City 20224, Taiwan; 3Center of Excellence for Ocean Engineering, National Taiwan Ocean University, Keelung City 20224, Taiwan

**Keywords:** Mesoporous silica nanoparticle, drug delivery system, target treatment, lanthanide metal, TAT peptide, hyaluronic acid, hyaluronidase

## Abstract

Nanoparticle-based drug delivery systems are among the most popular research topics in recent years. Compared with traditional drug carriers, mesoporous silica nanoparticles (MSN) offer modifiable surfaces, adjustable pore sizes and good biocompatibility. Nanoparticle-based drug delivery systems have become a research direction for many scientists. With the active target factionalized, scientists could deliver drug carriers into cancer cells successfully. However, drugs in cancer cells could elicit drug resistance and induce cell exocytosis. Thus, the drug cannot be delivered to its pharmacological location, such as the nucleus. Therefore, binding the cell membrane and the nuclear target on the nanomaterial so that the anticancer drug can be delivered to its pharmacological action site is our goal. In this study, MSN-EuGd was synthesized by doping Eu^3+^ and Gd^3+^ during the synthesis of MSN. The surface of the material was then connected to the TAT peptide as the nucleus target for targeting the cancer nucleus and then loaded with the anticancer drug camptothecin (CPT). Then, the surface of MSN-EuGd was bonded to the hyaluronic acid as an active target and gatekeeper. With this system, it is possible and desirable to achieve dual imaging and dual targeting, as well as to deliver drugs to the cell nucleus under a hyaluronidase-controlled release. The experimental approach is divided into three parts. First, we conferred the material with fluorescent and magnetic dual-imaging property by doping Eu^3+^ and Gd^3+^ into the MSN. Second, modification of the cell membrane target molecule and the nucleus target molecule occurred on the surface of the nanoparticle, making the nanoparticle a target drug carrier. Third, the loading of drug molecules into the carrier gave the entire carrier a specific target profile and enabled the ability to treat cancer. In this study, we investigated the basic properties of the drug carrier, including physical properties, chemical properties, and in vitro tests. The result showed that we have successfully designed a drug delivery system that recognizes normal cells and cancer cells and has good anticancer effects.

## 1. Introduction

Drug release systems based on nanoparticles have been widely used for cancer treatment. An effective drug release system needs to have sufficient drug loading capability and the ability to target to bring nanoparticles into the cancer cells preferentially [1]. However, drugs in cancer cells could elicit drug resistance and induce cell exocytosis. Thus, the drug cannot be delivered to its pharmacological location [2], such as the nucleus. Therefore, we will bind the cell membrane target and the nuclear target on the nanomaterial so that the anticancer drug can be delivered to its pharmacological action site and increase therapeutic efficiency.

A good drug carrier for a drug release system must have large drug loading efficiency, good biocompatibility, uniform size, and high stability. In recent years, many drug carriers have been developed [3]. Examples include liposomes [4], polymers [5], micelle [6], magnetic nanoparticles [7] and quantum dots [8]. Almost all nanoparticles are limited by instability and insufficient drug loading or toxicity and cannot be widely used. Mesoporous silica nanoparticles (MSN) as the carrier of the drug delivery system could overcome the previous disadvantage because: their high specific surface area allows MSN to modify more molecules on the surface [9]. Large and tunable pore volume can load more drug molecules [10], it has good biocompatibility, can be biodegraded and does not easily accumulate in the body [11]. The overall structure is composed of silica and Si-OH groups, which can provide a good environment to load and protect drugs and create many chemical surface modifications.

Most of the nanoparticle-based drug delivery systems enter the tumor tissue via the enhanced permeability and retention effect (EPR effect) [12], a postulate that nanoparticles, as well as molecules of certain size, are prone to accumulate in tumor tissue more than in normal tissue. To further enable nanoparticles to be effectively endocytosed by tumor cells, scientists will modify the active target on the surface of the nanoparticle [13,14]. Active targets are usually molecules that bind to receptors that are overexpressed on the surface of tumor cells compared to normal cells so that the nanoparticles can recognize the difference between normal cells and tumor cells.

However, successful entry of the nanoparticles through the cell membrane does not guarantee that the drug can be smoothly delivered to the desired pharmacological site. The drug carriers may be re-extracted out of a cancer cell via exocytosis, resulting in the insufficient concentration of the drug in the cell and reducing the cytotoxic effect. To solve this problem, scientists have simultaneously modified the cell membrane target and its drug-acting organelle target on the surface of the drug carrier. After the drug carrier enters the cell by endocytosis, the organelle target can lead the drug carrier to its targeting organelle [15].

Hyaluronic acid (HA) is the target selected for this experiment, and it is one of the main components of the extracellular matrix, which plays an important role in cell proliferation and migration [16]. Because cancer need to perform a large amount of proliferation and migration, hyaluronic acid receptors (CD44 receptor) are expressed to an excessive degree on the cancer cell surface [17], and the drug carrier can enter the cell by endocytosis through the binding of HA and CD44 receptor.

The nucleus is an important storage space for genetic material and plays an important role in the processes of cell metabolism, growth, and differentiation. Some anticancer drugs such as doxorubicin (DOX) or camptothecin (CPT) [18] induce cell apoptosis through drug entry into the nucleus, so it is very important to ensure that the drugs can enter the nucleus. For the drug carrier to pass through the nuclear membrane, it is necessary to interact with the nuclear pore complexes (NPC) on the nuclear membrane through a protein target which contains a nuclear localization signal (NLS) to allow the carrier to enter the nucleus [19]. TAT peptides [20], like other nuclear targets, such as dexamethasone (DEX) [21], are common nuclear targets. In this study, besides modifying HA, we will further modify the NLS contained TAT peptide, which can transport the drug carrier to the nucleus for drug release.

In addition to carrying the drug to the pharmacological site through the target on the surface of the drug carrier, we must have a gatekeeper to keep the drug in the drug carrier pore so that the drug does not release prematurely. Controlled release systems are mainly divided into external stimuli response and intrinsic stimuli response. External stimuli response is to make the gatekeeper decompose or structurally change by light or magnetic stimulation and then release the drug [22]. The intrinsic response is to use the difference between the internal and external environment of the cell, such as the change in pH or the difference in enzyme concentration, the gatekeeper can break down and release the drug after entering the cell due to environmental changes [23,24,25]. In this study, HA is not only used as an active target but also as a gatekeeper because of its polymer properties. Hyaluronidase (HAase) is an enzyme that catalyzes the hydrolysis of HA. There are six types of HAase in the human body [26], of which type I and type II are the primary enzymes that hydrolyze HA in the majority of tissues. Type II is mainly linked to the CD44 receptor and is responsible for cleaving the HA of the polymer, then further hydrolyzing the HA into the cell via endocytosis by type II [27]. Studies have shown that cancer cells use hyaluronidase to hydrolysis hyaluronic acid into smaller molecular fragments and elicit significant angiogenic effect [28]. When the drug carrier enters the cancer tissue and penetrates into the cell through endocytosis, the gatekeeper collapses due to the action of HAase, thereby achieving the purpose of releasing the drug into the cell [18].

According to previous laboratory research [29], two kinds of lanthanide metals with fluorescence [30] and magnetic imaging [31] functions, Eu^3+^ and Gd^3+^, are added to the synthetic process of MSN. The nuclear penetrating peptide (TAT peptide (sequence: YGRKKRRQRRR)) as a nuclear target was attached to the surface of MSN, then the anti-cancer drug (CPT) was loaded into the pore. Finally, the hyaluronic acid (HA) is used to attach to the surface of MSN as cell membrane target and gatekeeper. When the nanoparticles enter the cancer cells, the HA is decomposed by the HAase in the lysosome, and the nuclear target TAT is exposed, introduced nanoparticle into the nucleus for drug release. The study combines three functions of dual imaging with a controlled release switch and dual targeted treatment so that the material can simultaneously manifest the controlled release effect and increase the accumulation of drugs within cancer tissues. Finally, the imaging function is used to track the lesion location in clinical application (Scheme 1).

## 2. Materials and Methods

### 2.1. Materials

Tetraethyl orthosilicate (TEOS), hexadecyltrimethylammonium bromide (CTAB), 1-ethyl-3-(3-dimethylaminopropyl) carbodiimide (EDC), N-hydroxysuccinimide (NHS), hyaluronic acid sodium salt from *Streptococcus equi* (HA, mol wt: ~1.5–1.8 × 10^6^ Da), (3-aminopropyl) triethoxysilane (APTES), hyaluronidase from bovine test: Type I-S (HAase), camptothecin (CPT), and thiazolyl blue tetrazolium bromide (MTT) were purchased from Sigma-Aldrich (St. Louis, MI, USA). Sodium hydroxide (NaOH), toluene, and dimethyl sulfoxide (DMSO) were purchased from J.T.Baker and the N-acetyl TAT peptide (YGRKKRRQRRR) was synthesized by ^@^GenMark company (Carlsbad, CA, USA). Minimum essential media (MEM), F-12K (Kaighn’s) medium, fetal bovine serum (FBS), and antibiotic-antimycotic (AA) were purchased from Gibco (Waltham, MA, USA).

### 2.2. Synthesis of Eu(NO_3_)_3_ and Gd(NO_3_)_3_

Here, 4.40 g and 4.53 g of Eu_2_O_3_ and Gd_2_O_3_ were mixed with 4.89 mL and 5.03 mL of 16 M HNO_3_, respectively, and then hydrothermally heated at 180 °C for 24 h, after which the mixed mixture was added to the D.I. water to obtain 50 mL of 0.5 M Eu(NO_3_)_3_ and Gd(NO_3_)_3_.

### 2.3. Synthesis of MSN-EuGd-NH_2_

Ninety-seven milliliters of deionized water was added into 1.4 mL of 1 M NaOH and 0.2 g of surfactant CTAB. After stirring at 80 °C for one hour, 1 mL of TEOS and 3 mL of 0.5 M Eu(NO_3_)_3_, Gd(NO_3_)_3_ were added dropwise and stirred for two hours. The substances were washed with water, ethanol, and methanol and then calcinated at 650 °C for six hours to generate MSN-EuGd. Next, 0.1 g of MSN-EuGd was added to 15 mL of toluene and 0.2 mL of APTES, and it was stirred at 120 °C for four hours, centrifuged, and washed twice with alcohol to obtain MSN-EuGd-NH_2_.

### 2.4. MSN-EuGd-NH_2_ loaded into CPT (MSN-EGd-NH_2_@CPT)

10 mg of CPT was dissolved in 5 mL of dimethyl sulfoxide (DMSO), and 50 mg of MSN-EuGd-NH_2_ was added and mixed with ultrasonic waves for one hour. It was then stirred for 24 h, centrifuged three times and wash with deionized water to remove the most of solvent, then dried under vacuum for 48 h.

### 2.5. Synthesis of MSN-EuGd-TAT (or MSN-EuGd@CPT-TAT)

10 mg of N-acetylated TAT peptide was dissolved in 10 mL phosphate-buffered saline (PBS) (0.2 M, pH 7.4), and then, 9.6 mg EDC and 5.8 mg NHS were added at room temperature for half an hour. Next, 40 mL PBS (0.2 M, 80 mg of MSN-EuGd-NH_2_ (or MSN-EuGd-NH_2_@CPT) at pH 7.4) was dissolved, stirred for 12 h, centrifuged to remove the supernatant and lyophilized.

### 2.6. Synthesis of MSN-EuGd-TAT-HA (or MSN-EuGd@CPT-TAT-HA)

10 mg of MSN-EuGd-TAT (or MSN-EuGd@CPT-TAT) were dissolved in 4 mL MES solution (0.01 M, pH 5.5), 10 mg HA, 10 mg EDC and 10 mg NHS were added. After stirring at room temperature for 12 h, the supernatant liquid was removed by centrifugation and lyophilized.

### 2.7. Characterization

X-ray powder diffraction (XRD) was performed using a Bruker D2 Phase instrument. Particle size and zeta potential analyses were performed using dynamic light scattering (Malvern Zetasizer Nano ZS system, Malvern, Worcestershire, UK). Transmission electron microscopy images and energy dispersive X-ray (EDX) spectra were taken using a Tecnai F30 instrument. The analysis of nitrogen adsorption isotherms was performed using a Barrett–Joyner–Halenda (BJH) analysis (ASAP 2020, Micromeritics, Norcross, GA, USA). The surface area and pore size distribution curves of the undoped or various-doped mesoporous silica nanoparticles were determined by the Brunauer–Emmett–Teller (BET) method. The Fourier transform infrared (FTIR) spectra of the functionalized MSNs were recorded by using a BRUKER TENSOR Series II Spectrometer (Billerica, MA, USA). The luminescence excitation spectra were recorded using a Jasco FP-6300 photoluminescence spectrophotometer (Easton, MD, USA) with an excitation wavelength of 394 nm. The thermal gravimetric analysis (TGA) curves were obtained using a Netzsch TG 209 F3 apparatus to determine the conjugation efficiency of the TAT and HA when the temperature was increased to 800 °C. The drug release curve of the camptothecin was analyzed using an Enzyme-Linked Immunosorbent Assay (ELISA) reader (BioTek Synergy Mx, Winooski, VT, USA) at 430 nm. The T1-weighted magnetic resonance (MR) imaging was performed using conventional spin-echo acquisition (TR/TE = 300 ms/10.6 ms, slice thickness = 2.00 mm) using a 7 T scanner (BRUKER S300 BIOSPEC/MEDSPEC MRI, Karlsruhe, Germany). The concentrations of Eu^3+^ and Gd^3+^ ions that were doped into the MSNs were measured by inductively coupled plasma AES spectrometry (ICP-MS, Santa Clara, CA, USA) and reported as mass percentages.

### 2.8. Drug Release

10 mg of MSN-EuGd@CPT-HA was first stirred with 150 U/mL of 3 mL of HAase/PBS aqueous solution for 12 h, then the supernatant was removed by centrifugation and dried under vacuum. Next, the HAase-treated MSN-EuGd@CPT-HA and the HAase-untreated MSN@CPT-HA were compressed into bracts, placed in 3 mL of DMSO and shaken evenly, and 100 μL of the supernatant was aspirated into the 96-well disk at 10, 20, 30, 40, 50, 60, 80, 100, 120, 150, 180, 210, 240, 270, 300, 360, 420, and 480 min. The ELISA reader then measured the optical density (OD) value at 430 nm, and the total drug release amount was calculated according to the concentration calibration curve of the OD value of 430 nm previously read by ELISA.

### 2.9. In Vitro Experiments

#### 2.9.1. Cell Culture

L929 (Mus musculus fibroblast cell line) was cultured in minimum essential medium (MEM) supplemented with 10% fetal bovine serum (FBS) and 1% antibiotics (AA) at 37 °C in an environment containing 5% CO_2_, A549 (adenocarcinomic human alveolar basal epithelial cells) was cultured in F-12K supplemented with 10% FBS and 1% antibiotics (AA) at 37 °C in an environment containing 5% CO_2_.

#### 2.9.2. Cell Viability Assay

Normal cell model L929 and cancer cell model A549 were selected as test cells in this experiment. The procedures were as follows:

First, we seed 10,000 cells/well of cells in a 96-well culture dish and incubate the cell for 24 h in a 37 °C cell culture incubator. Then we add 25/50/100/200 μg/mL of drug carrier/culture solution in each well respectively. Next, after co-culture with the drug carrier for 24 h, 20 μL of MTT was added into the wells for four hours’ reaction. Finally, after the reaction, we add 100 μL of DMSO into each well and shake the dish for 15 min to induce its color. By reading the OD value at 540 nm with an enzyme immunoassay analyzer (ELISA reader, Winooski, VT, USA), the ability of cells reducing MTT can be known and can be used as an indicator of cell viability. The cell viability is calculated by the following formula:

Cell viability = OD540 (test group)/OD540 (control group) × 100%


#### 2.9.3. Confocal Image Analysis

The sterilized 13 mm glass coverslip was placed in a 24-well plate. Then, 2 × 10^4^ cells were seeded in each well, cultured for 24 h (5% CO_2_, 37 °C), and then cultured with a 500 μL (100 μg/mL) mixture of the drug carrier and the culture solution for six hours. After the completion of the culture, the culture medium was washed with PBS, and then 300 μL of a 3.7% formaldehyde/PBS solution was added for 10 min to fix the current state of the cells. After the end of the reaction time, the cells were washed with PBS, and then 4′,6-diamidino-2-phenylindole (DAPI) was used to stain the nuclei for five minutes. After washing with PBS, the coverslips were mounted onto a glass slide, and the cells were visualized and observed under a confocal laser-scanning microscope (CLSM, SP5, Leica, Wetzlar, Germany).

## 3. Results and Discussion

### 3.1. Structure, Formation, Morphology, and Properties of MSNs and EuGd-MSNs

Figure 1a shows the results of the low-angle XRD pattern. Both MSN and MSN-EuGd have characteristic peaks at (100) (110) (200), indicating that they are in the form of MCM-41 with regular hexagonal pore structure [33]. It can be seen that the structure of MSN-EuGd is similar to that of MSN, and MSN-EuGd does not cause a large change in structure due to the doping of Eu and Gd, its structural arrangement is similar to MSN. The MSN d-spacing was calculated by XRD pattern to be 3.68 nm, and the MSN-EuGd d_100_-spacing was 3.99 nm (Table 1). These results indicate that the MSN pore structure will change through the doping of metal ions, but this does not affect the main structure of MSN. The experiment uses BET analysis of MSN and MSN-EuGd. From Figure 1b nitrogen constant temperature adsorption and the pore size distribution pattern, it can be seen that the curve is of type IV and that all structures have a mesoporous structure as determined by hysteresis loop. MSN properties can be known by BET model calculation. The pore diameter of MSN-EuGd is 2.75 nm (Appendix A). After analysis, the specific surface area of MSN-EuGd is 608.19 m^2^/g, it is much larger than non-porous silica nanoparticle compared with the previous research [34], and the pore volume is 0.93 cm^3^/g (Table 1). The structure and size can be observed by TEM analysis. MSN has a regular hexagonal hole structure, and each particle has a uniform size. From Figure 1c–d, the hole size is approximately 2~3 nm as determined by XRD and BET. The measured data is consistent, and the particle size is approximately 120 nm. All of these geometric parameters are summarized in Appendix A. The DLS can transmit the laser light through the solution containing the nanoparticles, and the receiver receives the light and is affected by the particles to generate a scattering signal to calculate the hydration radius of the particles. It can be seen from Table 1 that the size of MSN-EuGd is 271 nm, and the size of the organic molecule can be changed as it is grafted onto the material. The surface charge of the nanoparticles is also measured. Confirming that each molecule connected to MSN-EuGd: The surface charge of MSN-EuGd is −14.5 mV, and the potential rises to −10 mV due to its positive charge when connected to -NH_2_ [35]. After the TAT peptide was attached, the potential was raised to 4.08 because the TAT peptide itself was positively charged [36]. Regarding HA attachment, the potential reached −17.3 because the HA itself was rich in negatively charged -COOH [37].

The EDX can be used to determine the elements contained in the material. From Appendix A, it can be found that elements such as silicon, oxygen, europium, and gadolinium are detected in MSN-EuGd, while MSN is only silicon (Si), oxygen (O), and then further quantified by inductively coupled plasma mass spectrometry (ICP-MS) to obtain Eu and Gd contents of 4.91% and 4.82%, respectively, as shown in Appendix A.

The MSN-EuGd measurement by PL found that: if 394 nm is used as the excitation wavelength, it will produce radiation absorption peaks at 590 nm and 615 nm, primarily from the red light emission peak of Eu^3+^ from 5D_0_→7F_1_ (590 nm) and 5D_0_→7F_2_ (614 nm) after receiving excitation light [38]. If a radiation wavelength of 615 nm is used, an absorption peak is observed at 394 nm as shown in Figure 2a, and Eu^3+^ is indeed dopeddop into MSN. In addition, if the material is irradiated with 254 nm UV light, MSN-EuGd will emit red excitation light, as shown in Appendix A. We used IVIS to illuminate MSN-EuGd at 430 nm excitation wavelength. From Figure 2b, MSN and blank were observed to have no obvious fluorescence characteristics, while MSN-EuGd showed very obvious fluorescence excitation, confirming that IVIS can effectively detect materials. The nature of the fluorescent light also confirms that the MSN-EuGd can use the IVIS system as an imaging tracking function.

For the magnetic properties of the material, we synthesized MSN-EuGd with different ratios of lanthanides and measured the results with a superconducting quantum interference device. It was found that the undoped Gd^3+^ material showed no magnetic properties, but MSN-EuGd doped with Gd^3+^ exhibits a paramagnetic phenomenon. As the concentration of Gd^3+^ escalates, the paramagnetic property is more pronounced, confirming that the material is paramagnetic (Figure 2c) [39]. MSN-EuGd can also be applied to MR imaging to perform in vitro MRI testing. The parameters are set in a magnetic field of 7 T, setting the parameter value TR/TE = 300 ms, FOV = 7 cm, NEX = 1, slice thickness = 2.00, matrix = 256 × 256, and material concentration from 4 to 0.25 mg/mL^3^. From Figure 2d, it can be seen that the T1-weighted image appears increasingly bright as the material concentration increases, confirming that MSN-EuGd can be used as the T1 positive development image [40].

To confirm whether the organic molecule was successfully attached to MSN-EuGd, we can use the FTIR to analyze the functional groups on the material (Appendix A). The -OH group was observed at 3400 cm^−1^ and 2931 cm^−1^, and the Si-O-Si signal [41] at 1068 cm^−1^ and 953 cm^−1^. When the -NH_2_ was modified, it was found that an additional N-H bond peak [42] on the amine group at 1552 cm^−1^ confirmed that the amino group was successfully attached to MSN-EuGd. Then, when TAT was modified, it was observed that 1415 cm^−1^ and 1715 cm^−1^, respectively, represent the C-O-H stretching vibration of the amide bond and the stretching vibration of C = O, which proved that TAT was successfully connected to MSN-EuGd. MSN-EuGd-TAT-HA showed an additional peak at 1409 cm^−1^, representing the asymmetric stretching vibration of the -COOH group of HA [43]. The amount of MSN-EuGd modified by organic molecules and its drug loading were determined by TGA analysis. In this experiment, the temperature of each material is increased to 800 degrees from 40 degrees Celsius in the environment of pure oxygen, and the mass loss percentage of each material is observed. Finally, the amount of each molecule connected to MSN-EuGd is converted into Appendix A. The modification amount of -NH_2_, -TAT and HA is approximately 129.17 mg/g, 26.27 mg/g, and 65.53 mg/g, respectively. The loading amount of CPT is 15.22 mg/g.

To confirm that MSN-EuGd@CPT-TAT-HA can be utilized for drug release, we first reacted MSN-EuGd@CPT-HA with 150 U/mL HAase for 12 h, then centrifuged to remove the supernatant liquid and added the residual material into DMSO for drug release test. As a result, it was found that the amount of MSN@CPT-HA that was not treated with HAase was approximately 44.86%, and the material treated with HAase was approximately 83.57% at eight hours. It can be seen that the design of this experiment can achieve the purpose of drug-controlled release, as shown in Figure 3.

### 3.2. In Vitro Cytotoxicity and Cellular Uptake of Functionalized MSN-EuGd

To confirm the phagocytosis between cells for each material, mouse fibroblasts (L929) were compared with human lung adenocarcinoma cells (A549) as a CD44 receptor control group, and CLSM images were taken after six hours of coculture with each material. As shown in Figure 4, the bare MSN-EuGd is barely phagocytized by L929 cells and A549 cells, while the MSN-EuGd-TAT demonstrates a slight overlap of the material signal (red) and the nuclear signal (blue). It is speculated that MSN-EuGd-TAT can successfully enter the nucleus by TAT after being phagocytized. The MSN-EuGd-HA also exhibited that A549 cells contained more phagocytic material than did L929 cells. It is speculated that the overexpressed CD44 receptor on A549 cells enables MSN-EuGd-HA to be introduced into the cells by receptor-mediated endocytosis of A549 cells. The MSN-EuGd-TAT-HA can be found to be similar to MSN-EuGd-HA, but it can be observed that the signal of MSN-EuGd-TAT-HA overlaps with the nuclear signal of A549 cells. The material is successfully predicted by cell membrane target (HA), and nuclear target (TAT) enter the nucleus of cancer cells.

Cell viability tests confirm that the material is cytotoxic to cancer cells and less harmful to normal cells. As shown in Figure 5 and Figure 6. In order to avoid possible cytotoxic interference caused by excessive uptake of MSNs by the cells, therefore, referred to the results of Chou et al. in 2017 and we choose the concentration of 200 μg/mL as the highest dose [44]. The results obtained were that L929 and A549 cells had a cell viability of more than 80% when using a drug carrier without a loading drug, indicating that the material itself is not cytotoxic to the cells. However, it was observed in the A549 group that MSN-EuGd@CPT-TAT was slightly more toxic to cells than MSN-EuGd, and the reason was that MSN-EuGd@CPT-TAT was introduced into the nucleus after entering the cells. Different concentrations of MSN-EuGd@CPT-HA showed that MSN-EuGd@CPT-HA had a better cytotoxic effect on A549 compared with A549 and L929. It is speculated that the binding of the CD44 receptor on the HA and A549 cells causes the cells to increase drug phagocytosis. Finally, MSN-EuGd@CPT-TAT-HA exhibits better cytotoxic effects against A549 than does MSN-EuGd@CPT-HA. This is primarily because MSN-EuGd@CPT-TAT-HA enters the cancer tissue and then carries the drug into the cell and onward to the nucleus via cell membrane and nucleus targeting, thus furthering cytotoxicity.

## 4. Conclusions

This study demonstrates the successful synthesis of a novel drug delivery system (MSN-EuGd@CPT-TAT-HA) that possesses dual development and dual targets and controls the release of drugs into the nucleus. The system is used to overcome the side effects of chemotherapy and multiple drug resistance problems.

The experiment used MSN with Eu and Gd as the carrier and loaded the anticancer drug CPT. After attaching the nuclear target TAT, it was connected with the HA which functions as both cell membrane target and gatekeeper. A nanoparticle, MSN-EuGd@CPT-TAT-HA, with dual target and dual development and HAase as a release switch was synthesized as a drug delivery system. After the material enters the tumor via the cell membrane target HA, HA is decomposed by HAase in the cytosol to expose the nuclear target TAT on the surface of MSN-EuGd, and the remaining particle enters the nucleus by TAT to release the drug.

The material confirms that doped lanthanides Eu and Gd through the PL and SQUID provided the material with a fluorescent imaging and magnetic imaging function. It is also confirmed from the IVIS and MRI images that the cancer tissue distribution can be tracked in vitro. The drug release test also confirmed the sealing ability of HA and the release of the drug by HAase decomposition of HA.

It was confirmed from the CLSM image that the MSN-EuGd was attached to the cell membrane target HA and the nuclear target TAT. The material can be introduced into the cell by endocytosis via the expression receptor and then onward into the nucleus. The cell viability test showed that MSN-EuGd alone demonstrated excellent cytocompatibility. When the material is loaded with the drug, it can also obtain better cancer cell cytotoxic effects with the cell membrane target HA and the nuclear target TAT attached to MSN-EuGd. Hopefully, this intelligent drug carrier can successfully become a potential therapeutic material for cancer.

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
