# Peer review of "Hyaluronidase-Responsive Mesoporous Silica Nanoparticles with Dual-Imaging and Dual-Target Function"

_cancers, 2019, doi:10.3390/cancers11050697_

Round 1
Reviewer 1 Report
This manuscript describes the delivery of hyaluronidase-responsive camptothecin loaded mesoporous silica nanoparticles. The nanoparticles were also incorporated with Eu3+ and Gd3+ and surface is functionalised with TAT peptide. The delivery system was tested base don cell toxicity in L929 and A549 cells.
In general, the manuscript requires thorough revision in English, language, terminologies. The delivery system and idea has been reported in literature by several researcher previously. The delivery of camptothecin has also been reported. The delivery strategy looks similar to previously reported literature; however, could not find the exact formulation. This concept of dual targeting of both nucleus and cell membrane looks good.
The study lacks the study on in vivo effectiveness of delivery system. The administration route and actual application is unclear. The
Title: I suggest rephrasing the title “Hyaluronidase-Responsive Mesoporous Silica Nanoparticles with Dual-Imaging and Dual-Target Function”
Abstract: Please rewrite. It lacks the essence of manuscript. Avoid discussions. Please include background, objectives, methods, results, and conclusion in sequence, where labels of the same is not required.
Introduction and discussion lack explanation on hyaluronidases that how it HAses from cancer cells only can target the HA of the delivery system than normal cells”. I strongly suggest including the discussion and literature on probable administration route for delivery system. The discussion/literature on the toxicity of MSN nanoparticles can be included. Toxic Dose? Route vs dose?
Please include the data/result to ensure that the DMSO was completely removed from test samples and did not affect the results.
Please keep the terminologies and SI units uniform.
How much is the loading and entrapment efficiency of the camptothecin in delivery system? Please include.
Line 180, seeding is more appropriate word than incubating.
Materials and methods of 2.9 section are suggested to be rewritten. Use of cell culture terminologies are inappropriate.
Figure 2: Please improve the resolution. Correct the spelling of wavelength.
Figure 4 needs to be relabelled to indicate third column in each cell line is both blue and red
Please include the statistics such as significance level, p value, etc wherever applicable.
Please improve figure 5 resolution and presentation. The presenting all reactants along with final product is confusing than clarifying. The terminology ‘@CPT’ can be replaced with other suitable terminologies.
Author Response
Thank you very much for providing these insights, they have hit the nail on the head.
Due to these important suggestions, we had improved our manuscript. Correction for errors and explanations has been made after we received your helpful reply letter. We hope that the new version of our manuscript can clearly present our work. The parts we have corrected in the manuscript have been highlighted. In addition, we have re-examined and edited the English editing. The relevant proof is also added to the last page of the manuscript.
Point 1: The study lacks the study on in vivo effectiveness of delivery system. The administration route and actual application is unclear. The Title: I suggest rephrasing the title “Hyaluronidase-Responsive Mesoporous Silica Nanoparticles with Dual-Imaging and Dual-Target Function.”
Response 1: We appreciate for your suggestion. We think that the corrected title is more simple and appropriate. We have changed the title “Hyaluronidase-Responsive Mesoporous Silica Nanoparticles with Dual-Imaging and Dual-Target Function.” in the new manuscript.
Point 2: Abstract: Please rewrite. It lacks the essence of manuscript. Avoid discussions. Please include background, objectives, methods, results, and conclusion in sequence, where labels of the same is not required.
Response 2: Thanks for the advice, we had rewritten the abstract and completed the revised abstract in highlight.
Point 3: Introduction and discussion lack explanation on hyaluronidases that how it HAses from cancer cells only can target the HA of the delivery system than normal cells”. I strongly suggest including the discussion and literature on probable administration route for delivery system. The discussion/literature on the toxicity of MSN nanoparticles can be included. Toxic Dose? Route vs dose?
Response 3: Thanks for pointing out the incomplete part in the manuscript.
In order to make the description more complete, we also added a note about the hyaluronidases explanation in line 99.
‘’Studies have shown that cancer cells use hyaluronidase to hydrolysis hyaluronic acid into smaller molecular fragments and elicit significant angiogenic effec 1.’’
We had added references into line 316 in the manuscript.
’’ As shown in figure 5 and figure 6: In order to avoid possible cytotoxic interference caused by excessive uptake of MSNs by the cells, therefore, referred to the results of Chou et al. in 20172 and we choose the concentration of 200 μg/ml as the highest dose.’’
So that the discussion/literature on the toxicity of MSN nanoparticles can be included.
Point 4: Please include the data/result to ensure that the DMSO was completely removed from test samples and did not affect the results.
Response 4: Thank you for your attention and reminder. After the drug loading process is completed, we wash the sample 3 times with deionized water, then centrifuged and lyophilized it to remove DMSO.
Point 5: Please keep the terminologies and SI units uniform
Response 5: Thanks for your reminding, terminologies and SI units have been unified.
Point 6. How much is the loading and entrapment efficiency of the camptothecin in delivery system? Please include.
Response 6: This is a very good question. We had confirmed the loading efficiency of the camptothecin, it is 15.22 mg/g through TGA measurement (table S3 and figure S5). We were also added sentences to lines 284-286.
’’The modification amount of -NH2 , -TAT and HA is approximately 129.17 mg/g, 26.27 mg/g, and 65.53 mg/g, respectively. The loading amount of CPT is 15.22 mg/g.’’
Point 7. Line 180, seeding is more appropriate word than incubating.
Response 7: Thank you very much for your correction. I have changed the ‘’incubation’’ from line 191 into ‘’seeding’’.
Point 8: Materials and methods of 2.9 section are suggested to be rewritten. Use of cell culture terminologies are inappropriate.
Response 8: Thanks for the suggestion, the inappropriate terminologies and some confusing statement in section 2.9.2 and 2.9.3 had been fixed in line 188-208’’
2.9.2. Cell Viability Assay
Normal cell model L929 and cancer cell model A549 were selected as test cells in this experiment. The procedures were as follows:
First, we seed 10,000 cells/well of cells in a 96-well culture dish and incubate the cell for 24 hours in a 37 °C cell culture incubator. Then we add 25/50/100/200 μg/mL of drug carrier/culture solution in each well respectively. Next, after co-culture with the drug carrier for 24 hours, 20 μl of MTT were added into the wells for 4 hours’ reaction. Finally, after the reaction, we add 100 μl of DMSO into each well and shake the dish for 15 minutes to induce its color. By reading the OD value at 540 nm with an enzyme immunoassay analyzer (ELISA reader), the ability of cells reducing MTT can be known and can be used as an indicator of cell viability. The cell viability is calculated by the following formula:
Cell viability = OD540 (test group)/OD540 (control group) × 100%
2.9.3. Confocal Image Analysis
The sterilized 13 mm glass coverslip was placed in a 24-well plate. Then, 2x104 cells were seeded in each well, cultured for 24 hours (5% CO2, 37 °C), and then cultured with a 500 μl (100 μg/ml) mixture of the drug carrier and the culture solution for 6 hours. After the completion of the culture, the culture medium was washed with PBS, and then 300 μl of a 3.7% formaldehyde/PBS solution was added for 10 minutes to fix the current state of the cells. After the end of the reaction time, the cells were washed with PBS, and then 4',6-diamidino-2-phenylindole (DAPI) was used to stain the nuclei for 5 minutes. After washing with PBS, the coverslips were mounted onto a glass slide and the cells were visualized and observed under a confocal laser-scanning microscope (CLSM, SP5, Leica).)
Point 9: Figure 2: Please improve the resolution. Correct the spelling of wavelength.
Response 9: Thanks for your correction, I have labeled Fig 2. (a)(b)(c)(d) and corrected the ‘’wavelength’’ spelling error. Regarding to the resolution of the MRI, we are unable to present a better resolution due to instrument limitations.
Point 10. Figure 4 needs to be relabelled to indicate third column in each cell line is both blue and red
Response 10: Thank you for reminder, we have modified Figure 4 to make it clearer.
Point 11. Please include the statistics such as significance level, p value, etc wherever applicable.
Response 11: In order to make the data appearance clearer, we added Figure 6 in line 334 to compare the cytotoxic effects on normal cells and cancer cells by using different concentrations of final products, and found significant differences in the group with a material concentration of 200 ug/ml, the P value would be less than 0.01.
Point 12: Please improve figure 5 resolution and presentation. The presenting all reactants along with final product is confusing than clarifying. The terminology ‘@CPT’ can be replaced with other suitable terminologies.
Response 12: We want to use the symbol “@” to indicate the difference between the drug loaded and modified. Such representation is also presented in other literature, for example: He, Y. J. et al.3 and Hakeem, A. et al.4 research.
References:
1. Khegai, II, Neurohormonal Regulation of Tumor Growth. Russ. J. Genet. 2018, 54 (1), 36-44.
2. Chou, C. C.; Chen, W.; Hung, Y.; Mou, C. Y., Molecular Elucidation of Biological Response to Mesoporous Silica Nanoparticles in Vitro and in Vivo. ACS Appl. Mater. Interfaces 2017, 9 (27), 22235-22251.
3. He, Y. J.; Liang, S. Q.; Long, M. Q.; Xu, H., Mesoporous silica nanoparticles as potential carriers for enhanced drug solubility of paclitaxel. Mater. Sci. Eng. C-Mater. Biol. Appl. 2017, 78, 12-17.
4. Hakeem, A.; Zahid, F.; Zhan, G. T.; Yi, P.; Yang, H.; Gan, L.; Yang, X. L., Polyaspartic acid-anchored mesoporous silica nanoparticles for pH-responsive doxorubicin release. Int. J. Nanomed. 2018, 13, 1029-1040.

Reviewer 2 Report
I think the overall quality of this paper is good enough for the publication.
However, there are some minor concerns to be fixed, as listed below:
An important thing that is lacking in the present text is that benefits of using MSN is not shown by data, for instance, corresponding data for silica nanoparticles having the same particle size but no mesopores.
Scheme 1: What is the driving force for breaking membrane of the endosome? Add a brief explanation in the text.
Table 1 & Line 203-4:
Indicate the mean particle diameters of MSN and MSN-EuGd, other than their BJH desorption diameters.
The “d-spacing” must be specified with a Miller index of the crystal plane. Indicate an appropriate Miller index as subscript such as d(100)-spacing.
Figure 2 : Add (a), (b), (c), or (d) at top left of each data.
Minor remarks
Line 14: it have become → it has become
Line 41: micell magnetic nanoparticles → micelles(?), ….
Author Response
Thank you very much for approval and advice, we found that your questions and suggestions are very helpful. According to these important suggestions, we had improved our manuscript. We hope that the new version of our manuscript can clearly present our work. The parts we have corrected in the manuscript have been highlighted.
Point 1: An important thing that is lacking in the present text is that benefits of using MSN is not shown by data, for instance, corresponding data for silica nanoparticles having the same particle size but no mesopores.
Response 1: Thanks for your reminding, we had view the research from da Silva, C. R. et al. 1, then we found that the specific surface area of nonporous silica nano particle is 15m2/g, it is much smaller that mesoporous nano particles. This means that MSNs can have more functionalized modification than nonporous silica nano particles. We also added in line 221:
’’MSN properties can be known by BET model calculation. The pore diameter of MSN-EuGd is 2.75 nm (figure S1). After analysis, the specific surface area of MSN-EuGd is 608.19 m2/g, it is much larger than non-porous silica nanoparticle compared with the previous research 1.’’
Point 2: Scheme 1: What is the driving force for breaking membrane of the endosome? Add a brief explanation in the text.
Response 2: This is a very good suggestion:
The driving force for destroying the endosome is due to the hydrolysis of hyaluronic acid, which leads to the imbalance of the osmotic pressure of the endosome, thus triggering the proton sponge effect, so that the nanoparticles can escape from the inner body and further enter the nucleus2. We also added a note on line 116.
’’Then, after the HA is hydrolyzed by the HAase between the cell membrane and the endosome and caused the proton sponge effect2 to escape endosome, the exposed TAT peptide on MSN is used to deliver the MSN to the nucleus for drug release.’’
Point 3: Table 1 & Line 203-4:
Indicate the mean particle diameters of MSN and MSN-EuGd, other than their BJH desorption diameters.
The “d-spacing” must be specified with a Miller index of the crystal plane. Indicate an appropriate Miller index as subscript such as d(100)-spacing.
Response 3: Thanks for your comment, in line 216 we had changed d-spacing into d100-spacing and added the radius of MSN and MSN-EuGd in Table 1.
Point 4: Figure 2 : Add (a), (b), (c), or (d) at top left of each data.
Response 4: Thanks for your suggestion, I have labeled (a)(b)(c)(d) in Fig 2.
Point 5: Minor remarks
Line 14: it have become → it has become
Line 41: micell magnetic nanoparticles → micelles(?), …
Response 5: Thanks for notice, these grammar errors will be fixed in the main manuscript, line 15 and line 48.
References:
1. da Silva, C. R.; Wallau, M.; Urquieta-Gonzalez, E. A., Mesoporous carbons prepared by nano-casting with meso- or non-porous silica nanoparticles. J. Braz. Chem. Soc. 2006, 17 (6), 1170-1180.
2. Smith, S. A.; Selby, L. I.; Johnston, A. P. R.; Such, G. K., The Endosomal Escape of Nanoparticles: Toward More Efficient Cellular Delivery. Bioconjugate Chem. 2019, 30 (2), 263-272.

Round 2
Reviewer 1 Report
manuscript has been improved.